# Metabolic Reprogramming of Immune Cells in the Tumor Microenvironment

**DOI:** 10.3390/ijms252212223

**Published:** 2024-11-14

**Authors:** Jing Wang, Yuanli He, Feiming Hu, Chenchen Hu, Yuanjie Sun, Kun Yang, Shuya Yang

**Affiliations:** Department of Immunology, The Fourth Military Medical University, Xi’an 710032, China; wwangjing0405@163.com (J.W.); yuanli5832@163.com (Y.H.); 17778914060@163.com (F.H.); 18579121005@163.com (C.H.); syjfly@163.com (Y.S.)

**Keywords:** the tumor microenvironment, metabolic reprogramming, immune cells, immunotherapy

## Abstract

Metabolic reprogramming of immune cells within the tumor microenvironment (TME) plays a pivotal role in shaping tumor progression and responses to therapy. The intricate interplay between tumor cells and immune cells within this ecosystem influences their metabolic landscapes, thereby modulating the immune evasion tactics employed by tumors and the efficacy of immunotherapeutic interventions. This review delves into the metabolic reprogramming that occurs in tumor cells and a spectrum of immune cells, including T cells, macrophages, dendritic cells, and myeloid-derived suppressor cells (MDSCs), within the TME. The metabolic shifts in these cell types span alterations in glucose, lipid, and amino acid metabolism. Such metabolic reconfigurations can profoundly influence immune cell function and the mechanisms by which tumors evade immune surveillance. Gaining a comprehensive understanding of the metabolic reprogramming of immune cells in the TME is essential for devising novel cancer therapeutic strategies. By targeting the metabolic states of immune cells, it is possible to augment their anti-tumor activities, presenting new opportunities for immunotherapeutic approaches. These strategies hold promise for enhancing treatment outcomes and circumventing the emergence of drug resistance.

## 1. Introduction

The tumor microenvironment (TME) refers to the surrounding environment in which tumor cells exist. This includes various immune cell types, cancer-associated fibroblasts, endothelial cells, pericytes, and a diverse array of other tissue-resident cell types. These cells were once considered bystanders in tumor development; however, it is now understood that they play a crucial role in the pathogenesis of cancer. The cells and extracellular components within the TME interact with tumor cells to promote tumor proliferation and invasion while also reducing drug permeability [1]. Among them, the immune microenvironment, which consists of immune cells recruited and activated by tumor cells along with associated stromal components, is a crucial element of the tumor microenvironment [2]. The immune microenvironment plays a dual role in tumor development. In the early stages of tumor colonization or growth, local immune cells and their associated factors create an anti-tumorigenic inflammatory microenvironment that inhibits tumor progression. However, as the tumor advances, it can also cause the immune microenvironment to facilitate its progression and affect the response to therapy.

In the TME, all immune components are collectively referred to as the tumor immune microenvironment (TIME). This encompasses adaptive immune cells, including T cells and B cells, as well as various myeloid cells such as macrophages, neutrophils, monocytes, dendritic cells (DCs), mast cells, eosinophils, and myeloid-derived suppressor cells (MDSCs). Additionally, other innate lymphoid cells (ILCs) and innate-like T cells (ILTCs) are integral to this complex immune landscape [3]. ILCs, including natural killer (NK) cells and helper ILCs (termed ILC1s, ILC2s, and ILC3s) and lymphoid tissue-inducing (LTi) cells. ILTCs (used interchangeably) are composed of three key subsets: natural killer T (NKT) cells, mucosal-associated invariant T (MAIT) cells, and γδ T cells [4]. Most immune cells exhibit both anti-tumor and pro-tumor effects [5]. Tumor-derived cytokines, chemokines, and various metabolic conditions can influence the function of immune cells across a range of cancers [6]. Interactions between tumor cells and stromal components, including rapid tumor growth and vascular anisotropy, lead to a TME characterized by significant hypoxia, low pH, and elevated pressure [7]. The immunoinflammatory responses triggered by numerous growth factors, cytokines, and various proteolytic enzymes are highly active within the TME. These responses promote the development of an immunosuppressive TME, facilitating immune evasion by the tumor and severely restricting the effectiveness of immunotherapy.

Overlapping metabolic reprogramming of tumor cells and immune cells is considered one of the key determinants of the anti-tumor immune response in cancer [8]. During tumorigenesis and development, the metabolic network of tumor cells undergoes reprogramming to meet the demands of continuous, uncontrolled cellular proliferation and to gain a survival advantage in a complex microenvironment [9]. In this context of significant nutrient and energy depletion, tumor metabolism can limit the destruction of tumor cells by immune checkpoints [10]. Tumor cells and immune cells coexist in the TME and dynamically interact at the metabolic level. Tumor cells require substantial amounts of ATP, amino acids, lipids, and sugars to sustain cellular activity and proliferation. Conversely, when immune cells transition from a quiescent state to an activated and functional state, they must also reprogram their metabolic pathways to meet the demand for specific nutrients, exhibiting a metabolically elevated phenotype similar to that of tumor cells [11]. This energetic interaction between tumor and immune cells fosters metabolic competition within the tumor ecosystem, which limits nutrient availability and contributes to microenvironmental acidosis, thereby impairing immune cell function [12]. Metabolic reprogramming is essential for various types of immune cells to maintain both their own homeostasis and that of the organism. Immune cells face metabolic constraints, which inevitably affect their vitality and function. Additionally, impaired immune cell metabolism can act as a catalyst that promotes tumor cell escape. Increasingly, studies have indicated that immune cells undergo metabolic reprogramming during proliferation, differentiation, and the execution of effector functions, which is crucial for an effective immune response.

## 2. Metabolic Reprogramming of Tumor Cells

Metabolic reprogramming, which involves accelerating cell growth and proliferation by regulating energy metabolism, is a significant characteristic of tumors [13]. Tumor cells modify their metabolic patterns to meet their energy demands and to obtain metabolic precursors necessary for rapid proliferation, as well as to supply raw materials for the synthesis of various biomolecules. Changes in intracellular and extracellular metabolites that accompany cancer-associated metabolic reprogramming can profoundly impact gene expression, cell differentiation, and TME. During tumor progression, tumor cells continuously absorb nutrients to support their rapid proliferation. These processes significantly influence the characteristics of TME, including altered nutrient availability, hypoxia, and the production of immunosuppressive metabolites [14]. Tumor cells consume large amounts of glucose and release substantial quantities of lactate through enhanced aerobic glycolysis, resulting in decreased glucose levels and lactate accumulation within the TME. The combination of low oxygen supply due to aberrant vascularization and high oxygen consumption by tumor cells maintains a hypoxic and acidic environment, which facilitates angiogenesis, invasive metastasis, and therapeutic resistance in tumor cells [15,16]. Additionally, this environment decreases the local radiosensitivity of tumor cells [17]. Due to the unfavorable conditions of hypoxia and nutrient deficiency, tumor cells undergo metabolic modifications that engage alternative core metabolic processes, such as glutamine catabolism and fatty acid oxidation, to meet their energy and anabolic requirements. This is evidenced by a significant increase in the demand for arginine and glutamine by tumor cells, as well as the utilization of various mechanisms to acquire lipids and enhance lipid oxidation [18] (Figure 1). Additionally, tumor cells can produce a range of immunosuppressive metabolites, including adenosine, 2-hydroxyglutarate, kynurenine, lactate, and methylthioadenosine, which inhibit anti-tumor immunity and facilitate tumor progression [19] (Table 1).

## 3. T Cell Metabolic Reprogramming

There are multiple T cell infiltrations in TME. After recognizing tumor antigens and co-stimulatory signals, the T cell receptors of naïve lymphocytes become activated and proliferate into effector T cells (Teff), which then infiltrate the tumor area under the influence of chemokines [49]. Based on their immune functions, Teff are classified into cytotoxic T cells (CTL), helper T cells (Th), and regulatory T cells (Tregs). Among these, CTLs represent the cytotoxic phenotype of CD8+T cells and are responsible for tumor-killing functions, while Tregs diminish the activity of effector T cells and promote immunosuppression within the TME. Th1 cells activate macrophages by secreting cytokines such as interferon-γ (IFN-γ) and tumor necrosis factor-α (TNF-α), thereby enhancing their ability to phagocytose and eliminate pathogens. Additionally, IFN-γ secreted by Th1 cells activates natural killer (NK) cells, facilitates the destruction of tumor cells by these NK cells, and contributes to the generation and enhancement of tumor-specific cytotoxic T lymphocyte (CTL) responses [50]. Th2 cells may counteract the effects of Th1 cells. When a shift in the Th1/Th2 balance occurs in cancer patients, leading to a predominance of Th2 cells, this can mediate immune evasion by reducing antigen presentation [51]. Th17 cells secrete cytokines such as interleukin-17 (IL-17) and interleukin-21 (IL-21), which positively regulate the functions of Th1 and NK cells, thereby mediating anti-tumor immune responses. Furthermore, Th17 cells exhibit overlapping phenotypic characteristics with regulatory T (Treg) cells, and many of the chemotactic receptors on their surfaces are identical, suggesting that Th17 cells possess plasticity [52]. Malignant tumors alter the metabolic programs and functions of T cells through various strategies to sustain a suppressive tumor microenvironment. These strategies include impaired immune cell infiltration, T cell depletion, and the infiltration of suppressive immune cells.

### 3.1. Impaired Infiltration of Immune Cells

Tumor-infiltrating T lymphocytes (TILs) play a crucial role in both immune clearance and immune evasion within the TME. TILs must sustain their nutrient supply and energy requirements during proliferation, differentiation, and effector functions following antigenic stimulation. Tumor cells and TILs compete directly for essential nutrients, such as glucose and glutamine, in the suppressive TME. This competition hampers T cell metabolism and effector function, ultimately driving tumor progression. Based on the spatial distribution of cytotoxic T lymphocytes (CTLs) within the TME, tumors can be classified into three primary immune phenotypes: immune-inflammatory, immune-exclusion, and immune-desert phenotypes [53]. Immune-exclusion and immune-desert tumors are often referred to as immune-exempt TME [54], characterized by a limited presence of immune cells or suppressive subpopulations, such as Tregs, MDSCs, and tumor-associated macrophages (TAMs). In such instances, effector immune cells fail to efficiently penetrate the tumor microenvironment, instead remaining confined to the peripheral stroma, thereby limiting their capacity to exert a tumor-suppressive effect. The genesis of an immunosuppressive TME is attributed to the dysregulation of T cell activation, which is influenced by a multitude of factors. Among the key cellular actors that exert their effects through metabolic pathways are MDSCs. The heightened aerobic glycolysis in tumor cells can result in the upregulation of granulocyte-macrophage colony-stimulating factor (GM-CSF) and granulocyte colony-stimulating factor (G-CSF), which in turn stimulates the proliferation of MDSCs. This process further dampens T cell functionality and augments the immunosuppressive capabilities of the tumor [55]. MDSCs can competitively deplete cysteine within the extracellular milieu, thereby upregulating the activity of iNOS (inducible nitric oxide synthase) and Arg-1 (arginase-1). This leads to the consumption of L-arginine and consequently impedes T cell generation. Additionally, MDSCs can suppress T cell immune responses by generating reactive oxygen species (ROS). Co-culturing MDSCs with T cells reveals a significant suppression of T cell functionality and proliferation [56].

### 3.2. T Cell Depletion

A hallmark characteristic of TME is the depletion of T cells, which is often induced by malignant tumor cells or Tregs. The persistent antigenic stimulation and the presence of immunosuppressive factors within the TME lead to a gradual exhaustion of tumor-infiltrating CD8+ T cells, culminating in a loss of their effector cytotoxic and proliferative capabilities [57]. This exhaustion is exacerbated by the glucose scarcity in the TME, a consequence of the heightened glucose consumption by cancer cells, which subsequently inhibits the tumoricidal activity of TILs. Moreover, tumor cells that highly express checkpoint molecules, such as PD-1 and CTLA-4, disrupt the metabolic pathways of T cells. This disruption is achieved by inhibiting glucose transporter 1 (Glut1) and glycolysis while simultaneously promoting lipid oxidation. Other immune cells, including DCs, MDSCs, and TAMs, contribute to T cell depletion through the overexpression of enzymes like arginase and indoleamine 2,3-dioxygenase (IDO). This overexpression leads to the depletion of essential amino acids, such as arginine and tryptophan, further impairing the metabolic fitness of T cells and altering their activation and differentiation processes [58]. A multitude of immunosuppressive factors within the TME can impact T cell differentiation and precipitate T cell failure. These factors include an adverse cytokine milieu, nutritional deprivation, and exposure to immunosuppressive molecules. Tregs, which often accumulate in excess within the TME, express immunosuppressive molecules including IL-10, IL-35, and TGF-β, further contributing to T cell depletion. During interactions with Tregs, effector T cells may also experience glucose consumption, cellular senescence, and DNA damage. Furthermore, the TME impairs the mitochondrial biogenesis and function of peroxisome proliferator-activated receptor γ coactivator 1-α (PGC-1α) in TILs, which is crucial for their metabolic fitness and overall anti-tumor response [59].

In this section, we explore the aberrant metabolic processes that contribute to the depletion of CD8+ T cells within TME. Tumor cells within the TME exploit metabolic reprogramming, notably aerobic glycolysis, as a survival strategy. Specific microenvironmental stressors, such as hypoxia, nutrient depletion, and the accumulation of metabolic byproducts, can impede the metabolic activities of CD8+ T cells and disrupt their differentiation pathways. These factors culminate in the formation of functionally impaired CD8+ T cells, characterized by a progressive decline in their cytotoxic and proliferative capabilities, ultimately leading to their functional exhaustion [57]. The aberrant metabolic pathways that drive the differentiation of CD8+ T cells into exhausted phenotypes include the following:

#### 3.2.1. Reprogramming of Glucose Metabolism Induces Depletion of CD8+ T Cells

Tumor cells, through the Warburg effect, outcompete CD8+ T cells for glucose, leading to glucose scarcity that directly compromises the functional activity of CD8+ T cells and their capacity to produce IFN-γ. This metabolic competition not only hampers tumor clearance but also fosters tumor progression [60]. Furthermore, elevated levels of immune checkpoint molecules contribute to CD8+ T cell depletion [61]. PD-1 expressed on the surface of CD8+ T cells inhibits T cell glycolysis by suppressing the PI3K/Akt/mTOR signaling pathway. Conversely, PD-L1 on the surface of tumor cells can bind to PD-1 on CD8+ T cells, thereby inhibiting T cell function. Additionally, PD-L1 promotes the expression of glycolytic enzymes by facilitating the Akt/mTOR signaling pathway in tumor cells within the tumor microenvironment, leading to the translation of Glut1 [62], further intensifying tumor glycolysis. Consequently, the competitive suppression of glycolytic activity in TILs within the tumor results in the impairment of CD8+ T cell effector functions. Additionally, research indicates that early amelioration of the glucose-deprived environment for CD8+ T cells can still rejuvenate their functional efficacy. However, once CD8+ T cells experience prolonged nutritional deprivation, the production of inhibitory cytokines becomes relatively irreversible, leading to more permanent dysfunction that cannot be remedied simply by re-exposure to nutrients [63].

#### 3.2.2. Hypoxia Induces Depletion of CD8+ T Cells

The TME is often characterized by a persistent hypoxic state. While CD8+ T cells can still enter the effector phase following either continuous stimulation or hypoxia alone, the combination of these factors, particularly under conditions of sustained hypoxia, can expedite T cell depletion [64]. Research has established that this synergistic effect may stem from the impact on mitochondrial function [65]. Prolonged exposure to hypoxia and relentless antigenic stimulation within the TME can lead to the disruption of mitochondrial architecture and function. This disruption triggers the accumulation of CD8+ TILs with mitochondrial depolarization and aberrant production of mitochondrial reactive oxygen species (mtROS), culminating in intolerable ROS levels. Excessive ROS production has been demonstrated to further augment the transcriptional mechanisms of exhausted CD8+ T cells by inducing sustained NFAT signaling and acting as tyrosine phosphatase inhibitors [66]. This process manifests as functional, transcriptional, and epigenetic alterations characteristic of terminally exhausted CD8+ T cells, ultimately leading to severe functional impairments and the depletion of CD8+ T cells.

#### 3.2.3. Reprogramming of Lipid Metabolism Induces Depletion of CD8+ T Cells

The TME of highly proliferating tumor cells is rich in lipids. The main manifestations were increased de novo synthesis of fatty acids, accumulation of adipocytes, and lipoid fibroblasts. Unbalanced lipid metabolism is involved in T cell senescence. Gene expression of an important enzyme that synthesizes cholesterol was increased in aging T cells, and an increase in cholesterol was observed in TILs. In addition, the accumulated LDL contributes to the development of T cell senescence. CE, FFA, and cholesterol levels in senescent T cells were significantly increased. Studies have shown that cholesterol esterification increases the proliferation and function of T cells with impaired anti-tumor effects [67]. Accumulation of long-chain fatty acids in TME can inhibit the mitochondrial function of CD8+TILs in pancreatic ductal adenocarcinoma and trigger the major transcriptional reprogramming of the lipid metabolism pathway, thereby leading to the reduction in fatty acid catabolic metabolism and inducing CD8+TIL depletion, thus inhibiting T cell activity [68]. In addition, studies have shown that the high levels of cholesterol in the tumor microenvironment in vivo can induce an increase in CD36 expression of CD8+ T cells, which leads to excessive intake of fatty acids, lipid oxidative damage, and iron death, resulting in the loss of its lethal function and inducing it to enter a state of exhaustion [69,70]. Phospholipid metabolic reprogramming also leads to anti-tumor immunity. It was discovered that the reduced levels of phosphatidylcholine (PC) and phosphatidylethanolamine (PE) in CD8+ T cells within the tumor were attributed to the diminished expression of phosphatidylphosphatase 1 (Plpp1), an enzyme that catalyzes the synthesis of PE and PC [71]. T cell-specific deletion of Plpp1 impairs anti-tumor immunity and promotes T cell death caused by iron apoptosis. In addition, unsaturated fatty acids in TME can also stimulate iron apoptosis of Plpp1/CD8+ T cells.

#### 3.2.4. Reprogramming of Amino Acid Metabolism Induces Depletion of CD8+ T Cells

Arginine has been shown to greatly promote the activation and effector function of immune cells, especially T cells and NK cells [72]. The study found that L-arginine is strongly absorbed by activated CD8+ T cells in amounts that exceed the requirements for protein synthesis and can be rapidly converted into downstream metabolites by metabolic enzymes, such as those necessary for the production of ornithine for the synthesis of polyamines. Proteome and metabolome analyses showed that T cell metabolism was reprogrammed as L-arginine levels in CD8+ T cells increased. In particular, the key to promoting sugar dysplasia involves the upregulation of the enzyme pyruvate carboxylase (PC), the enol phosphate type pyruvate carboxylase (phosphoenolpyruvate carboxykinase, PCK2), and fructose 1,6-bisphosphatase (FBP1). Conversely, the levels of glucose transporters and glycolytic enzymes were found to be reduced. This means that an increase in arginine causes CD8+ T cells to consume less glucose, further shifting metabolism in activated T cells from glycolysis to mitochondrial oxidative phosphorylation. On the contrary, however, there are a lot of arginine metabolic enzymes in the tumor microenvironment (ARG Arginase-1, 1). Tumor cells will prefer to use their consumption of arginine, and most tumors lack argininosuccinate synthetase 1 (ASS1). It is a key enzyme for the production of arginine, resulting in the loss of intracellular arginine synthesis [73]. In this case, it encourages tumor cells to utilize exogenous arginine, which further reduces the availability of amino acids in CD8+ T cells and thus inhibits anti-tumor immunity [74]. Tryptophan also plays an important role in immune cells. T cell activation is extremely sensitive to the concentration of tryptophan in the surrounding environment. Tryptophan metabolism primarily involves the conversion of tryptophan into kynurenic acid through the action of two dioxygenases: indoleamine 2,3-dioxygenase (IDO) and tryptophan 2,3-dioxygenase (TDO). High levels of these two tryptophan-degrading enzymes are expressed in tumor cells [75], and tryptophan is heavily utilized by tumor cells in the tumor immune microenvironment, resulting in the lack of tryptophan leading to T cell apoptosis [76]. Glutamine is an intrinsically synthesized non-essential amino acid, but most tumor cells cannot survive in a medium lacking glutamine [77], and the dependence of tumor cells on glutamine is called “glutamine addiction” [78]. Glutamine is important for the rapid proliferation of CD8+ T cells, while glutamine deficiency enhances the depletion of CD8+ T cells [79]. This depletion is manifested by an increased expression of co-immunosuppressive molecules such as PD-1 and LAG-3. It was found that during T cell adoptive therapy, ammonia gradually accumulated in the adoptive CD8+ effector T cells in the tumor microenvironment. Inhibition of glutamine metabolism or overexpression of CPS1 to reduce ammonia levels in T cells can significantly improve the survival of T cells in the tumor microenvironment and enhance their anti-tumor effect [80].

### 3.3. Inhibitory Immune Cell Infiltration

As a member of the CD4+T cell family expressing FOXP3, Tregs are a subgroup of T cells with significant immunosuppressive effects [81]. The metabolic reprogramming of Tregs is also an important mechanism for their immunosuppressive function. By adjusting their metabolic pathway, Tregs can exert a stable immunosuppressive effect in extreme TME. Promote tumor cells to escape and kill other immune cells. Studies have demonstrated that CXCR3+ Treg cells are significantly enriched in human ovarian cancer, particularly in solid tumors, which contributes to immunosuppression [82]. It is noteworthy that Th17 cells have the ability to differentiate into Tregs, which subsequently exert an immunosuppressive effect [83].

#### 3.3.1. Glucose Metabolism

The low-glycemic environment of TME inhibits the PI3K-Akt-mTOR signaling pathway, leading to the down-regulation of the glycolysis pathway in CD4+ T cells. This causes them to shift to the fatty acid oxidation (FAO) and oxidative phosphorylation (OXPHOS) pathways to obtain energy and raw materials for biomolecule synthesis. In addition, CD4+ T cells are stimulated to express the nuclear factor FOXP3, which promotes the differentiation of Tregs [84]. However, reprogramming of glucose metabolism is often associated with destabilized immunosuppressive function of Tregs in mouse colorectal cancer models and human ovarian cancer [85,86]. In addition, lactate promotes PD-1 da in Tregs cells in tumors. In addition to activating CD8+ T cells in tumors, PD-1 blockade also activates Treg cells and enhances their inhibitory activity, thereby affecting the efficacy of immunotherapy [87].

#### 3.3.2. Lipid Metabolism

The lipid metabolism pathway of Tregs is very important for their immunosuppressive function. Fatty acid synthesis (FAS) promotes the functional maturation of Treg cells. FAO is the main energy source of Treg cells in TME, so Treg cells still play an immunosuppressive effect in the environment of glucose deficiency. FAS may play a leading role in regulating the proliferation of Tregs active in cancer and inflammatory conditions [88]. Studies have revealed that tumor-infiltrated Treg cells significantly up-regulate the expression of multiple fatty acid-binding proteins and CD36, which are responsible for the uptake of long-chain fatty acids and oxidized low-density lipoproteins [89]. Cd36-induced FFA uptake activates peroxisome proliferator activation receptor signaling, promotes mitochondrial adaptation, and increases the ratio of NAD+/NADH. Thus, Tregs have preferential survival and functional advantages in lactate-rich TME [90]. In addition, SREBP activity was up-regulated in Tregs. SREBP works in conjunction with SREBP cleavage-activating protein (SCAP) to regulate de novo fatty acid synthesis in Tregs by stimulating fatty acid synthase (FASN) [88].

#### 3.3.3. Amino Acid Metabolism

A number of studies have proved that when tumor cells transform glutamine, a large amount of glutamate is produced. This increase in glutamate levels promotes the expression of the metabolic type glutamate receptor 1 of Tregs, increases the uptake and utilization of glutamate, and promotes the proliferation and infiltration of Tregs [91]. IDO expression is upregulated in Tregs, which further catalyzes the conversion of tryptophan to kynurenine and induces FOXP3 expression in Tregs [92]. Kynurenine enhances the expression of FOXP3 by binding to and activating the cytoplasmic transcription factor aryl hydrocarbon receptor (AHR), thereby improving the amino acid uptake capacity of Tregs [93]. In metastatic melanoma, Treg up-regulates the arginine catalytic enzyme arginase 2, increases the uptake capacity of Arg, promotes its proliferation capacity and recruitment to tumor tissues, and inhibits anti-tumor immunity [94].

## 4. Metabolic Reprogramming of Tumor-Infiltrating Myeloid Cells

Tumor-infiltrating myeloid immune cells, including TAMs, MDSCs, tumor-associated neutrophils (TANs), etc., in an inhibitory immune microenvironment play a key role in forming and maintaining [95]. The rapid proliferation of cancer cells leads to the formation of a unique metabolic microenvironment that can significantly alter the phenotype and function of the infiltrating myeloid cells.

### 4.1. Macrophage Metabolic Reprogramming

In TME, macrophages are divided into two types based on the dynamic changes in TME at different times of tumor development. One type of macrophages, which appear early in tumor development and are activated by IFN-γ or LPS in TME, are polarized toward the M1-like phenotype and are called M1-like tumor-associated macrophages [96]. M1 macrophages have anti-angiogenic effects, which can promote chronic inflammation and inhibit tumor growth. The other type mainly occurs in the process of tumor proliferation and metastasis, and macrophages induced by IL-4 and IL-10 are polarized to the M2-like phenotype, which is called M2-like tumor-associated macrophages [97]. M2 macrophages express high levels of IL-10, TGF-β, ARG1, and other related factors that promote tumor growth, which can not only induce immune escape, angiogenesis, tumor growth, and metastasis but also lead to resistance to checkpoint inhibitors or adoptive T cell immunotherapy [98]. In general, what we call TAMs are manifested as M2 phenotypes that promote tumor progression.

Immunosuppressed M2 phenotype tumor-associated macrophages (TAMs) preferentially generate energy through oxidative metabolic pathways, such as oxidative phosphorylation (OXPHOS) and fatty acid oxidation (FAO). While M2 phenotype TAMs require a certain level of glycolysis to provide substrates and rapid energy for cytokine synthesis, their glycolytic activity is significantly lower than that of M1-like macrophages. Instead, M2 TAMs are primarily powered by the oxidative phosphorylation pathway, characterized by a high density of internal mitochondria and an increased oxygen consumption rate. Additionally, their tricarboxylic acid cycle is highly dependent on glutamine (Gln) intake [99]. Zinc finger E-box binding Homeobox 1 (Zeb1) is a transcription factor that can directly activate or inhibit gene expression by binding to regulatory regions of target genes. Studies have shown that ectopic Zeb1 directly increases the rate of glycolysis to determine the transcriptional expression of enzymes HK2, PFKP, and PKM2, thus promoting the Warburg effect and the proliferation, migration, and chemotherapy resistance of breast cancer in vivo and in vitro. In addition, Zeb1 exerts its biological effects by inducing glycolytic activity via the PI3K/Akt/HIF-1α signaling axis in response to hypoxia, which contributes to the cultivation of an immunosuppressive tumor microenvironment. Mechanically, breast cancer cells that ectopically express Zeb1 produce lactic acid in an acidic tumor environment and induce alternating activation of TAMs by stimulating the PKA/CREB signaling pathway [100]. Within TME, macrophages upregulate genes associated with ER stress response and increase the activity of the protein kinase RNA-like ER kinase (PERK) signaling cascade in macrophages. PERK activation promotes immunosuppressive M2 activation and proliferation through downstream transcription factor ATF-4 mediating up-regulation of phosphoserine aminotransferase 1 (PSAT1) and serine biosynthesis [41]. In addition, the accumulation of lactate and lipids in the TME under acidic conditions is crucial for the polarization of TAMs. TAMs uptake increased amounts of lipids from tumor cells by upregulating the expression of the scavenger receptor CD36. This process leads to a continuous accumulation of lipids within TAMs, enhancing fatty acid oxidation and oxidative phosphorylation. Furthermore, TAMs can be polarized into the M2 state through the activation of the STAT6 signaling pathway [101]. Arginine metabolism is a characteristic of macrophage polarization. A high level of Arg1 in M2 phenotype TAMs can hydrolyze arginine into ornithine and urea, while the decrease in arginine will affect the activation and proliferation of T cells and NK cells, thus triggering immune suppression [102]. In addition, studies have shown that the metabolic characteristics of TAMs are associated with colorectal cancer and are achieved through ATGL coactivator 5 (ABHD531), which is required during the breakdown of triacylglycerol [103]. ABHD5 is highly expressed in tumor-associated macrophages (TAMs) in colorectal cancer, where it inhibits the self-expression of spermine synthase through the C/EBPε signaling pathway. This inhibition leads to a reduced production of spermidine, thereby eliminating the inhibitory effect of spermidine derived from TAMs on the proliferation of colorectal cancer cells. Increased glycolysis in tumor-associated macrophages (TAMs) contributes to tumor angiogenesis, while the inhibition of the mechanistic target of rapamycin (mTOR) suppresses this effect [104]. This underscores the significance of macrophage metabolism in promoting tumor growth. Furthermore, TAMs metabolize cholesterol and locally produce 27-hydroxycholesterol (27-HC) in breast tissue. 27-HC accumulates in tumor tissues and induces macrophages to secrete CCL2, CCL3, CCL4, and other chemokines to promote monocyte recruitment and differentiation to M2 type. The interaction between 27-HC and M2-type TAMs further promotes the progression of breast cancer [105]. In conclusion, macrophages in TME are polarized, and metabolic reprogramming changes occur in differentiated TAM during the stage of tumor development, thus enabling macrophages to adapt to TME and promoting tumor deterioration.

### 4.2. Metabolic Reprogramming of Dendritic Cells (DCs)

Metabolic reprogramming of DCs within the tumor microenvironment has received limited attention. Recent studies indicate that DCs in this environment exhibit an immunosuppressive metabolic state. Notably, alterations in cholesterol metabolism and the associated mevalonate (MVA) metabolic signals in DCs can significantly influence immune responses [106,107]. The MVA pathway is a crucial component of cholesterol metabolism. DCs in the tumor microenvironment demonstrate a highly active MVA pathway, which activates small GTPases with the assistance of geranylgeranyl diphosphate (GGPP), a downstream metabolite of the MVA pathway. Small GTP enzymes play a crucial role in intracellular antigen transport pathways. Overactivation of small GTPases leads to accelerated antigen transport to lysosomes, which in turn reduces dendritic cell (DC) antigen presentation, impairs the initiation of cytotoxic T lymphocytes, and ultimately contributes to uncontrolled tumor progression [108]. Furthermore, tumor-derived cytokines can promote the accumulation of triacylglycerols, cholesterol esters, and fatty acids in DCs, thereby diminishing the expression of MHC-I on these cells and obstructing the presentation of exogenous antigens [109].

Accumulating evidence suggests that various tumor types are associated with the accumulation of tolerant DC populations [110]. The alteration of DC metabolism is a significant driving factor in tumor-mediated immune escape. Tumor cells and tumor-associated stromal cells produce high levels of lactic acid through aerobic glycolysis, which has been shown to inhibit DC interleukin-12 (IL-12) expression and impair DC-dependent in vitro antigen presentation [111]. In addition to creating an acidic microenvironment, developing tumors also generate hypoxic regions that promote the local expression of hypoxia-inducible factor-1 (HIF-1α). HIF-1α has recently been identified as a crucial contributor to DC-mediated regulatory T cell (Treg) differentiation in the intestinal mucosa and as an inhibitor of DC-dependent type 1 T helper cell (Th1) polarization in atherosclerosis. Tumor-associated hypoxia also promotes the accumulation of adenosine in tumor tissues and organs. Adenosine induces a tolerance phenotype in tumor-infiltrating dendritic cells (DCs) and activates adenosine receptors, which leads to an increased production of immunosuppressive and tolerance factors by the DCs. These factors include interleukin-10 (IL-10), cyclooxygenase-2 (COX-2), indoleamine 2,3-dioxygenase 1 (IDO1), arginase 1 (ARG1), arginase 2 (ARG2), prothrombin, and vascular endothelial growth factor (VEGF) [24]. These substances hinder Th1 polarization and impair the activation of cytolytic CD8+ T cells, ultimately contributing to tumor progression. Furthermore, studies have demonstrated that adenosine can inhibit the immunostimulatory properties of DCs by activating the adenosine A2B receptor (A2BAR) [112]. All of these alterations are expected to impede Th1 polarization and impair the activation of cytolytic CD8+ T cells. In terms of lipid metabolism, fatty acid synthase (FAS) is essential for the maturation of DCs. Research has shown that alpha-fetoprotein secreted by hepatocellular carcinoma cells can inhibit the normal immune function of DCs by suppressing their FAS activity through the inhibition of sterol regulatory element-binding protein 1 (SREBP-1) and peroxisome proliferator-activated receptor gamma coactivator 1-alpha (PGC1-α) in DCs [113]. In the context of amino acid metabolism, type 1 classical dendritic cells and cancer cells compete for the nutrient glutamine. Glutamine plays a crucial role in mediating antigen presentation by type 1 classical dendritic cells and activating CD8+ T cells through the FLCN-TFEB signaling axis. Consequently, cancer cells can evade immune responses from dendritic cells by monopolizing glutamine [114].

### 4.3. Neutrophil Metabolic Reprogramming

A significant number of neutrophils infiltrate the tumor microenvironment. Within this environment, neutrophils are categorized into N1 and N2 subtypes, as well as tumor-associated neutrophils (TANs) and inhibitory lobulated nuclear sources of myeloid cells, known as polymorphonuclear myeloid-derived suppressor cells (PMN-MDSCs) [115]. Notably, many of the pro-tumor activities exhibited by TANs are similar to those of PMN-MDSCs [116]. Studies have shown that both human and mouse PMN-MDSCs specifically upregulate fatty acid transporter 2 (FATP2), which facilitates the utilization of arachidonic acid and the synthesis of prostaglandin E2, thereby mediating immunosuppressive functions [117]. Furthermore, neutrophils can be categorized into three distinct states—T1, T2, and T3—based on their location within the TME. This categorization results from the epigenetic reprogramming of these cells [118]. Among them, increased chromatin accessibility associated with genes upregulated by T3 neutrophils (referred to as T3 genes) includes genes linked to hypoxia, glycolysis, and angiogenesis, such as *Vegfa* and *Hk2*. This accessibility facilitates pathways for cellular stress and survival, encompassing responses to hypoxia, oxidative stress, and glycolysis [119]. Consequently, T3 neutrophils are more prevalent in regions with elevated scores for glycolysis, hypoxia, and angiogenesis. Additionally, T3 neutrophils are abundant in genes related to angiogenesis, including *Vegfa*, *Thbs1*, and *Lgals3*. The interplay of these genes indicates a significant pro-tumor effect.

### 4.4. Metabolic Reprogramming of MDSCs

As the precursor cells of dendritic cells, macrophages, and granulocytes, myeloid-derived suppressor cells (MDSCs) exhibit strong immunosuppressive functions that inhibit the activity of T cells, NK cells, and dendritic cells. This inhibition promotes tumor development, metastasis, and drug resistance [120]. In colorectal cancer, the glycolytic capacity of MDSC precursor cells (immature myeloid cells, IMC) is significantly increased; however, glucose-deficient culture conditions do not impair their ability to expand and produce functional MDSCs. This phenomenon is attributed to an enhanced compensatory level of glutamine metabolism in IMC. The degradation of glutamine produces α-ketoglutaric acid and glutamate, which respectively promote the expansion and immunosuppressive functions of MDSCs [121]. In addition, MDSCs play a crucial role in remodeling the liver cancer microenvironment [122]. In liver cancer, tumor cells promote the recruitment and expansion of MDSCs within the TIME by releasing hypoxia-inducible factor 1-alpha (HIF-1α) and tumor-associated cytokines, such as IL-6, IL-1β, GM-CSF, G-CSF, VEGF, MCP-1, and MIF [123,124]. The expression of inducible nitric oxide synthase (iNOS) and Arg1 in MDSCs increases significantly, which can directly inhibit the anti-tumor functions of T cells. Additionally, PD-L1 on the surface of MDSCs can further suppress the anti-tumor response of T cells by binding to PD-1 on T cells. In a mouse model of hepatocellular carcinoma, TGF-β1 secreted by MDSCs significantly reduced the expression of NKG2D and the secretion of IFN-γ by NK cells. Neutralizing TGF-β1 restored the effector functions of NK cells. Furthermore, the immunosuppressive cytokines IL-10 and TGF-β1 secreted by MDSCs promoted the expansion of regulatory T cells, further inhibiting the effector functions of NK cells and CD8+ T cells [125]. In terms of the lipid metabolism, it has surfaced that a high-fat diet and fat induced the accumulation of somatic MDSCs in hormonal mice, and MDSCs promoted the rapid growth and spontaneous metastasis of primary tumors by inhibiting the activation of CD8+ T cells [126]. Cholesterol metabolism generates 27-HC, which promotes the differentiation of monocyte-derived myeloid-derived suppressor cells (M-MDSCs). This process increases the abundance of intratumoral M-MDSCs, contributing to the formation of an immunosuppressive tumor microenvironment and thereby facilitating tumor progression [127,128].

## 5. Summary and Prospect

With a deeper understanding of the mechanisms of the immune system, immunotherapy is introducing innovative approaches to cancer treatment. Over the past decade, tumor immunotherapy has transformed the landscape of cancer care. However, the complexity and heterogeneity of the TME contribute to its dynamic evolution through compensatory feedback mechanisms, the obstruction of immunotherapeutic effects, the development of drug resistance, and even tumor progression. Current tumor immunotherapies, including immune checkpoint inhibitors (ICBs), adoptive cell transfer therapy (ACT), and other modalities, often exhibit limited efficacy and may lead to drug resistance. Typically, these therapies are effective in only a small percentage of patients. The primary factors that hinder the effectiveness of immunotherapy include the low responsiveness of the host immune system to tumor antigens, the limited infiltration of immune cells in solid tumors, and the establishment of an immunosuppressive TME [129]. Therefore, comprehending the immune metabolism of T cells and other immune cells, as well as the metabolic behavior of cancer cells, is crucial for designing new cancer therapies. Targeting the metabolism of the TME can rectify metabolic disorders induced by tumor progression and improve the metabolic capacity of immune cells. When combined with other immunotherapeutic strategies, this approach represents a novel method for tumor treatment. Through an in-depth study of the overlapping metabolic reprogramming between immune cells and tumor cells within the tumor microenvironment, it is possible to regulate the balance of tumor metabolism and immune metabolism. This regulation can alter the metabolic pathways of immune cells, thereby enhancing their anti-tumor capabilities. For instance, by modulating the metabolic state of T cells, we can promote their differentiation into effector T cells instead of regulatory T cells, which may improve the effectiveness of immunotherapy and provide potential strategies for targeting metabolic pathways in the context of anti-cancer immunotherapy. A number of combination therapies targeting tumor metabolism and immune checkpoint inhibitors are currently undergoing preclinical or clinical trials, which will shed light on tumor immunotherapy. In various digestive tumors, including gastric, esophageal, colorectal, and pancreatic cancers, the high expression of lactate dehydrogenase A (LDHA)—a key enzyme in the aerobic glycolysis pathway of tumor cells—is often associated with poor prognosis and a high metastatic rate. Consequently, the research and development of anti-tumor drugs targeting LDHA have garnered significant attention. Oxamate, a pyruvate-competitive LDHA inhibitor, has been shown to inhibit the proliferation of gastric cancer cells; however, in vivo experiments could not be conducted due to its high effective dose [130]. Given the complexity and multilevel nature of the TME, the limited clinical application of immunotherapy targeting specific molecules or signaling pathways is a significant challenge. There remains considerable potential for improvement in this area. This review summarizes the metabolic reprogramming of immune cells in the tumor immune microenvironment and examines how these changes influence anti-tumor immune responses. In addition to the various types of immune cells discussed in this paper, there are many other significant immune cells within the tumor microenvironment that are subject to complex metabolic regulation. It is precisely due to the complexity and heterogeneity of the tumor microenvironment that the TIME has emerged as a crucial target for tumor immunotherapy, garnering increasing attention. Specific mechanisms and therapeutic strategies for regulating the metabolism of various nutrients in tumor cells, as well as for influencing the TIME to enhance the immune response, require further exploration. Future directions in tumor immunotherapy should focus on designing more targeted therapeutic approaches, developing personalized biomarker profiles, and improving efficacy while reducing the toxicity of combination drug therapies. The future direction of tumor immunotherapy involves designing more targeted treatments, developing personalized biomarker profiles, and further enhancing the efficacy while reducing the toxicity of combination drug therapies. Additionally, efforts are being made to decrease the incidence and recurrence rates associated with immunoprophylactic strategies [131]. Although challenges remain, this field of immunotherapy research is undoubtedly promising.

It is important to note that immune cell metabolites play a significant role in the complex effects of metabolic reprogramming within the TIME, influencing tumor development and suppression. These metabolites are involved in immune responses through various receptors, transporters, and post-translational modifications. In recent years, researchers have investigated strategies to monitor different immune cell metabolites as potential biomarkers for immunotherapy. This includes molecules associated with T-cell exhaustion [132,133,134], such as PD-1, TIM-3, and LAG3; markers of abnormal glucose metabolism for tracking and targeting dendritic cells, such as azide-modified mannose [135]; and molecules closely linked to the immunosuppressive functions of myeloid-derived suppressor cells [56], including Arg1, iNOS, and reactive oxygen species (ROS). The selection and application of these markers can provide critical information for monitoring and predicting responses to cancer treatment, thereby facilitating personalized medicine and enhancing treatment efficacy.

## Figures and Tables

**Figure 1 ijms-25-12223-f001:**
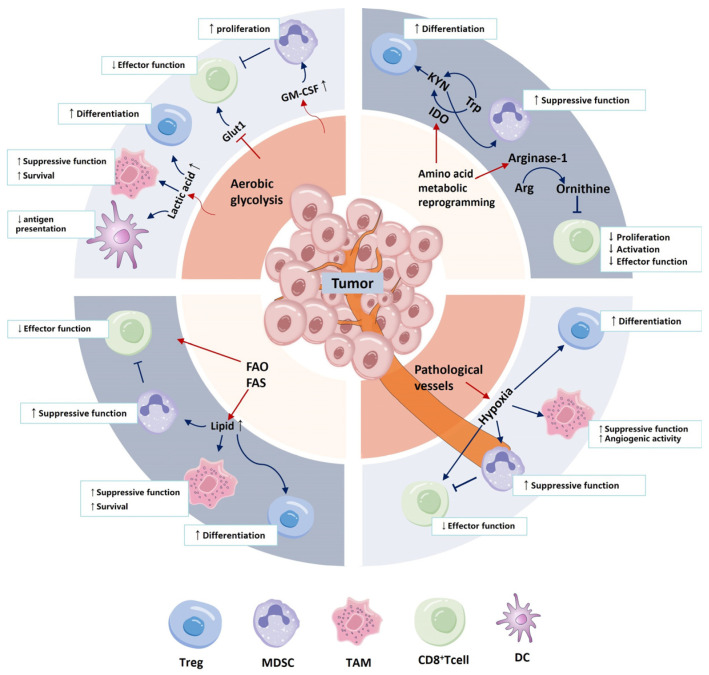
Overlapping metabolic reprogramming of tumor cells and immune cells. An upward arrow signifies that this metabolic reprogramming process leads to promotion, whereas a downward arrow indicates inhibition.

**Table 1 ijms-25-12223-t001:** Different inhibitory effects of various immunosuppressive metabolites on immune cells.

Immunosuppressive Metabolites	Immune Cell Types	Roles	Refs
Adenosine	NK	Blocking the maturation and migration of NK cells to tumor sites, as well as inhibiting their effector functions.	[20,21,22]
	DCs	Inhibit the antigen-presenting function of dendritic cells (DCs), regulate their differentiation and maturation, and suppress the anti-tumor immune response.	[23,24,25]
	CD4+ T cells	Inhibit the proliferation and differentiation of Th1 and Th2 cells, thereby suppressing both cellular and humoral immunity.	[26,27]
	CD8+ T cells	Induces depletion of CD8+ T cells.	[28,29]
	TAMs	Promotes the polarization, proliferation, migration, and invasion of M2-like macrophages (tumor-promoting macrophages).	[30,31]
	MDSCs	Promotes survival, proliferation, and immunosuppressive functions.	[32]
	Tregs	Promotes cellular proliferation, Foxp3 expression, and immunosuppressive functions.	[33,34]
2-hydroxyglutaric acid	CD8+ T cells	Inhibits cell proliferation and cytokine production, while reducing cytotoxicity.	[35]
Kynurenine	NK, APC(DC, macrophages)	Suppresses immune activity and inhibits differentiation.	[36,37]
	CD4+ T cells	Blocks proliferation and induces apoptosis in Th1 cells.	[38,39]
	Tregs	Promote Proliferation and Differentiation	[40]
Lactic acid	TAMs	Promotes the polarization of TAMs to the M2 type.	[41]
	DCs	Reduce antigen presentation and inhibit cytokine production and activation.	[24]
	NK	Inhibits the cytolytic function of natural killer (NK) cells, leading to reduced NK cytotoxicity, and is typically associated with decreased expression of perforin and granzyme in these cells.	[42]
Methylthioadenosine	Tregs	Promotes differentiation while maintaining suppressed immune function.	[43,44]
	DCs	Impedes DC migration and affects the initiation of the immune response.	[45]
	NK	Promotes anti-tumor immunity.	[46]
	MDSCs	Promotes migration, recruitment, activation, and immunosuppressive functions.	[47,48]

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
