# Peer review of "Metabolic Reprogramming of Immune Cells in the Tumor Microenvironment"

_ijms, 2024, doi:10.3390/ijms252212223_

Round 1

Reviewer 1 Report

Comments and Suggestions for Authors

Manuscript entitled "Metabolic Reprogramming of Immune Cells in the Tumor Microenvironment" by Jing Wang, et al. This manuscript provides an in-depth review of the metabolic reprogramming of immune cells within the tumor microenvironment (TME) and how these shifts influence tumor progression and responses to immunotherapy. The topic is highly relevant to ongoing research in tumor immunology, and the authors provide a thorough examination of the interplay between tumor cells and immune cells, as well as the metabolic adaptations that occur in both.

Comments:

1. The manuscript would benefit from a more detailed exploration of specific immune cell phenotypes, particularly the various subsets of T lymphocytes and innate immune cells. For example, while the paper touches on the roles of cytotoxic CD8+ T cells and regulatory T cells (Tregs), a deeper dive into the specific metabolic profiles of other key T cell subsets, such as Th1, Th2, and Th17 cells, would provide a more comprehensive picture of T cell function in the TME. Similarly, expanding the discussion on the role of natural killer (NK) cells and innate lymphoid cells (ILCs) in modulating anti-tumor immunity would be valuable.

2. A key missing element is a discussion on the dominant immune response phenotype in the TME. The authors should clarify whether a particular phenotype, such as a Th2-dominated or Treg-dominated immune response, typically prevails in different cancer types or stages of tumor development. 

3. The review would benefit from an explicit discussion of knowledge gaps in the current understanding of metabolic reprogramming within the TME. Additionally, the future directions section could be expanded to suggest new avenues for research, such as how combination therapies*ć targeting both immune cell metabolism and traditional cancer pathways might improve treatment efficacy and overcome drug resistance.

4 The manuscript should include more concrete clinical recommendations based on the findings. For instance, the authors could propose strategies for monitoring metabolic markers in immune cells as potential biomarkers for therapy response.

Author Response

Comments 1: The manuscript would benefit from a more detailed exploration of specific immune cell phenotypes, particularly the various subsets of T lymphocytes and innate immune cells. For example, while the paper touches on the roles of cytotoxic CD8+ T cells and regulatory T cells (Tregs), a deeper dive into the specific metabolic profiles of other key T cell subsets, such as Th1, Th2, and Th17 cells, would provide a more comprehensive picture of T cell function in the TME. Similarly, expanding the discussion on the role of natural killer (NK) cells and innate lymphoid cells (ILCs) in modulating anti-tumor immunity would be valuable.

Response 1: Thank you for pointing this out. We agree with this comment. Therefore, We have expanded our discussion on the roles of Th1, Th2, and Th17 subsets within the TME, highlighting their distinct functions in tumor immunity and immune evasion. Additionally, we have included a detailed analysis of the interplay between these T cell subsets and natural killer (NK) cells. This addition aims to provide a more comprehensive understanding of the multifaceted roles of T cells within the TME. We believe this enhancement aligns with your expertise and will be of significant interest to the readers of the journal. This change can be found - page 4, paragraph 1, line 118-130.

Comments 2: A key missing element is a discussion on the dominant immune response phenotype in the TME. The authors should clarify whether a particular phenotype, such as a Th2-dominated or Treg-dominated immune response, typically prevails in different cancer types or stages of tumor development. 

Response 2: Thank you for pointing this out. We agree with this comment. In response to your suggestion, we have added a section in the "3.3 Inhibitory Immune Cell Infiltration" part of our manuscript that elucidates the predominant role of Tregs in mediating immunosuppression within ovarian cancer. We have included the latest findings and their implications for therapeutic intervention, which we hope will enrich the scientific discourse on this critical aspect of tumor immunology. This change can be found - page 8, paragraph 1, line 294-298

Comments 3: The review would benefit from an explicit discussion of knowledge gaps in the current understanding of metabolic reprogramming within the TME. Additionally, the future directions section could be expanded to suggest new avenues for research, such as how combination therapies*ć targeting both immune cell metabolism and traditional cancer pathways might improve treatment efficacy and overcome drug resistance.

 Response 3: Thank you for pointing this out. We agree with this comment. Therefore, in the "Summary and Prospect" section, we have addressed the current gaps in knowledge regarding metabolic reprogramming within the TME. We have also discussed the state of research and ongoing clinical trials involving the combination of metabolic inhibitors and immune checkpoint inhibitors. This update is intended to provide a cutting-edge overview of this rapidly evolving field and its potential impact on cancer therapy. This change can be found - page 12, paragraph 2, line 507-516, 518-521, 529-538, 548-553

Comments 4: The manuscript should include more concrete clinical recommendations based on the findings. For instance, the authors could propose strategies for monitoring metabolic markers in immune cells as potential biomarkers for therapy response.

Response 4: Thank you for pointing this out. We agree with this comment. Therefore, We have supplemented the "Conclusion and Perspectives" section with a discussion on strategies for monitoring immune cell metabolic markers as potential biomarkers for therapeutic response. This addition underscores the clinical relevance of our research and its potential to inform future treatment strategies. This change can be found - page 13, paragraph 2, line 559-571

Reviewer 2 Report

Comments and Suggestions for Authors

The aim of this review article is to describe the metabolic reprogramming (MR) of TME immune cells and how this contributes to tumorigenesis, progression, and antitumor immunity. 
The listing of cellular components of innate and adaptive immunity in the Introduction section between L41-48 is unclear and requires revision.
In chapter 2, the authors write: This paper focuses on the metabolic reprogramming processes in tumor cells and their effects on the TME. The title and article, which present the metabolic reprogramming of immunocompetent cells, contradict this statement. I recommend rephrasing or removing the sentence.
The MR of T cells is described in detail among the elements of the TME. This section is in order. 
The MR of myeloid cells is discussed in the next chapter. The descriptions provided here are also accurate.
However, they do not describe the MR of B cells, which is also important for the anti-tumor and protumor effects of TME.
Figures and tables are provided to aid understanding of the text. 
The bibliography is adequate. 
The English language is understandable; I found only minor errors in the text.
We should correct the aforementioned flaws and complete the article. I suggest a major revision. 

Author Response

Comments 1: The listing of cellular components of innate and adaptive immunity in the Introduction section between L41-48 is unclear and requires revision.

Response 1: Thank you for pointing this out. We agree with this comment. We have revised the introduction section, specifically lines 42-49, to provide a more precise and detailed delineation of the cellular components of innate and adaptive immunity. This revision aims to clarify any ambiguity regarding the roles of these cells in the context of our study, ensuring that our readers have a clear understanding of the immune landscape we are investigating. This change can be found - page 1, paragraph 2, line 42-49

Comments 2: In chapter 2, the authors write: This paper focuses on the metabolic reprogramming processes in tumor cells and their effects on the TME. The title and article, which present the metabolic reprogramming of immunocompetent cells, contradict this statement. I recommend rephrasing or removing the sentence.
Response 2: Thank you for pointing this out. We agree with this comment. In response to your concern about potential ambiguity, we have removed the sentence "This paper focuses on the metabolic reprogramming processes in tumor cells and their effects on the TME." from Chapter 2. This deletion is intended to prevent any misinterpretation of the scope and focus of our research. This change can be found - page 2, paragraph 3, line 86-89

Comments 3: However, they do not describe the MR of B cells, which is also important for the anti-tumor and protumor effects of TME.
Response 3: Thank you for pointing this out. Our manuscript primarily discusses the metabolic reprogramming of T cells and myeloid cells within the tumor micro-environment. We concur with your assessment that B cells play an indispensable role in tumorigenesis and the regulation of antitumor immunity. We appreciate your recognition of this and are eager to explore the significance of B cells in future submissions to your esteemed journal, should the opportunity arise.

Comments 4: The English language is understandable; I found only minor errors in the text.
Response 4: Thank you for pointing this out. We agree with this comment. We have meticulously reviewed the manuscript and corrected all linguistic errors to ensure that the text is academically rigorous and professionally presented. This revision enhances the overall quality and readability of our work.

Round 2

Reviewer 1 Report

Comments and Suggestions for Authors

I am pleased to inform you that the authors have satisfactorily addressed all my comments and concerns. The manuscript has been significantly improved and now meets the required standards for publication. Therefore, I recommend that the manuscript be accepted for publication

Reviewer 2 Report

Comments and Suggestions for Authors

I accept the answers of the authors, the revised version is now can be published.